# Peer review of "Irradiation Behaviors in BCC Multi-Component Alloys with Different Lattice Distortions"

_metals, doi:10.3390/met11050706_

Round 1

Reviewer 1 Report

This is a nice study on the properties under irradiation of some bcc-multicomponent alloys.

The paper is clearly written, the figures carefully presented help the understanding of experimental results, the methodology correct and well explained and therefore I agree with the publication of this manuscript in Metals in a form close to the present one.

The only thing that I would like to be improved is the discussion, which I feel somewhat incomplete. Indeed, the paper presents some results, that the are important and confirm some oprevious results. But I would like tosee some discussion of how these results could lead tot he development of the field, either as possible applications, or the production of new alloys (or about some theory which could explain the experimental data).

Author Response

Thanks for your careful reading and constructive suggestions.

I have added several paragraphs at the end of Section 3.2. which concretely explain the reason why the material possessing a lower vacancy formation energy has slow void swelling. And the possible enlightenment for future material’s design is mentioned in the last paragraph of Section 3.2.

Thanks for your kind help again.

Best regards,

Yue Su, Peking University

Reviewer 2 Report

The authors study irradiation resistance of Fe, CrFeV, and AlCrFeV.

There is significant room for improvement in this manuscript.

My first and foremost suggestion is that the authors should classy these alloys as medium or low entropy alloys since the authors are using HEA-related concepts throughout the paper.

More specific comments are as under:

  1. Abstract:

Please mention the reason why/how Al improves the irradiation resistance of

 AlCrFeV. Also, why  CrFeV shows higher irradiation resistance than alpha-Fe.

  1. Introduction:

Since authors’ composition don’t belong to HEA and Ni-containing equiatomic alloys, I suggest that the discussion of these materials should be removed from the introduction part.

Line 42-46: It is required to extend the literature survey, as the irradiation resistance of many other BCC multicomponent alloys have been reported. The following papers should be cited:

  1. Kareer, J.C. Waite, B. Li, A. Couet, D.E.J. Armstrong, A.J. Wilkinson, Short communication: “Low activation, refractory, high entropy alloys for nuclear applications,” J. Nucl. Mater. 526 (2019) 0–5. doi:10.1016/j.jnucmat.2019.151744.

Waseem, Owais Ahmed, and Ho Jin Ryu. "Helium ions irradiation analysis of W0. 5 (TaTiVCr) 0.5 for application as a future fusion plasma-facing material." Materials Chemistry and Physics 260 (2021): 124198.

Waseem, Owais Ahmed, et al. "Effects of F3+ ion implantation on the properties of W and W0. 5 (TaTiVCr) 0.5 for depth marker-based plasma erosion analysis." Nuclear Materials and Energy 25 (2020): 100806.

  1. Materials and Methods

Line 69: The significance of choosing Au ions and 500 oC irradiation temperature should be described briefly.

Please mention how long did the irradiation take? How were the samples cooled after irradiation?

  1. Results and Discussions

XRD: why CrFeV shows very high intensity of (110) diffraction peak? Please mention the reason.

Also, please add the grazing incidence XRD of irradiated samples.

Figure 2(a) shows intergranular and transgranular regions. The chemical composition of both of these regions should be reported, maybe using point EDS, or any other technique.

Figure 2(b) is missing.

Line 102-119:  The section describing thermodynamic parameters is not properly organized. This may be re-organized. All the equations should be written in separate lines with the equation numbers.

Check typing error in line 115.

Line 102-119: Please write your statement in the paper on “how these thermodynamic parameters and empirical correlation, which were developed for/from High-Entropy Alloys (5-13 elements, 5-35at.%) are applicable to your compositions? Which are not high-entropy alloys.”

Line 127:  Please mention the reason why “At the fluence of 1.5×1016 cm−2 at 500 ℃, the 127 irradiated voids are our primary objects of observation”?

Line 131-132: 300 voids in how big area? Better to report voids per unit area.

Line 152: Figure 3(d-f) and (g-1) are missing.

Line 191-192: Because this is the main reason why AlCrFeV shows better irradiation resistance, I suggest the author explain this statement a little more.

Line 204: Please explain what does ‘hardening rate’ mean here? How was it calculated? Is it a percentage increase in hardness or something else?

Line 211: please avoid repetition.

Line 220: are there any ‘defect clusters’ in AlCrFeV?

How many times were the indentation tests repeated? Is it possible that the softening is just an experimental error? Did the authors try Nanoindentiaon at various regions of interest?

Line 228-229: please mention ions, energy, and dose value.

Author Response

Thanks for your patient reading and nice suggestions.

Round 2

Reviewer 2 Report

The authors have attended all the comments and made amendments accordingly. The revised paper can be accepted for publication.